# LEARNING TO AUGMENT INFLUENTIAL DATA

## ABSTRACT

Data augmentation is a technique to reduce overfitting and to improve generalization by increasing the number of labeled data samples by performing label preserving transformations; however, it is currently conducted in a trial and error manner. A composition of predefined transformations, such as rotation, scaling and cropping, is performed on training samples, and its effect on performance over test samples can only be empirically evaluated and cannot be predicted. This paper considers an influence function which predicts how generalization is affected by a particular augmented training sample in terms of validation loss. The influence function provides an approximation of the change in validation loss without comparing the performance which includes and excludes the sample in the training process. A differentiable augmentation model that generalizes the conventional composition of predefined transformations is also proposed. The differentiable augmentation model and reformulation of the influence function allow the parameters of the augmented model to be directly updated by backpropagation to minimize the validation loss. The experimental results show that the proposed method provides better generalization over conventional data augmentation methods.

## 1 INTRODUCTION

In supervised learning, deep neural networks generally require large amounts of labeled data for training. Insufficient labeled data will lead to poor generalization due to overfitting. One simple method to reduce overfitting and to improve generalization is to perform data augmentation, which involves creating additional labeled data by transforming each training sample with label preserving transformations. Different augmentation can bring about huge differences in test performance depending on the task (Dosovitskiy et al. (2016); Graham (2014); Cui et al. (2015); Zhang et al. (2015)). Unfortunately, it is an area which has not been extensively explored. Even on a well-studied image classification task (Simonyan & Zisserman (2015); He et al. (2016); Xie et al. (2017)), training data is often augmented in a manner similar to that performed in training the AlexNet (Krizhevsky et al. (2012)). A composition of predefined transformations, such as rotation, translation, cropping, scaling and color perturbation, is a popular choice; however, determining the strength of each transformation that will lead to the best performance is an art and is often set in an empirical manner with trial and error by observing validation loss (Simard et al. (2003); Ciregan et al. (2012)).

An alternative to tuning a composition of predefined transformations is to learn a strategy for composing the transformations (Sixt et al. (2018); Ratner et al. (2017); Lemley et al. (2017); Peng et al. (2018); Cubuk et al. (2018)). To learn a strategy, various criteria have been considered. Sixt et al. (2018) and Ratner et al. (2017) considered a strategy for augmenting realistic samples without considering the classification model. Given a classification model, Lemley et al. (2017) and Peng et al. (2018) considered antithetical strategies for augmenting samples that respectively minimize (Lemley et al. (2017)) or maximize (Peng et al. (2018)) training loss. It is not fully understood why these antithetical studies that relate data augmentation to training loss are effective in improving performance on test samples. Cubuk et al. (2018) considered small child models to compute a validation loss for augmenting samples to improve generalization. Here, learning requires a reinforcement learning framework with the validation loss as a reward signal. It is not clear how well the validation loss of the small child model approximates the validation loss of the final classification model which may have little relevance to the child model.

This paper proposes a data augmentation method that is able to directly trace the impact of the augmentation model on generalization. For this, we considered an influence function (Cook & Weisberg

(1980); Koh & Liang (2017)) to compute how the validation loss is affected by a particular augmented training sample. The influence function approximates the change in validation loss due to inclusion or exclusion of the augmented training sample without a leave-one-out retraining process. We also propose a differentiable augmentation model that generalizes the conventional augmentation method by the composition of predefined transformations. Reformulation of the influence function and the differentiable augmentation model enable a gradient of the validation loss to flow through the augmented samples and augmentation model; thus, the augmentation model can learn to augment samples to directly improve generalization by backpropagation.

The remainder of this paper is organized as follows. Section 2 briefly reviews some of the most relevant studies in the literature related to the proposed method, and Sections 3 and 4 describe the details of the proposed method. The experimental and comparative results are reported in Section 5. The conclusions are presented in Section 6.

## 2  RELATED WORK

### 2.1  DATA AUGMENTATION METHODS

In this subsection, we provide a brief overview of data augmentation methods in the following three categories: (i) unsupervised methods that do not consider the classification model during learning[1], (ii) adversarial methods that maximize classification loss and (iii) supervised methods that minimize classification loss.

**Unsupervised methods**  Unsupervised methods include the conventional data augmentation method, which is a composition of predefined transformations, such as rotating, translating, cropping, scaling and color perturbation (Simonyan & Zisserman (2015); He et al. (2016); Xie et al. (2017); Krizhevsky et al. (2012)). The transformations are manually chosen in an empirical way with trial and error by observing the validation loss (Simard et al. (2003); Ciregan et al. (2012)). Ratner et al. (2017) considered a generator that generates a sequence of predefined transformations. Given the sequence, the training sample is augmented by applying predefined transformations in a consecutive way. The generator is learned in the generative adversarial network (GAN) framework (Goodfellow et al. (2014)) that the generated sequence produces a realistic augmented sample; however, the classifier is not considered during the learning process and the effect of data augmentation can only be observed by trial and error.

**Adversarial methods**  Adversarial methods include hard sample mining that collects or augments samples that are misclassified by the current classification model. It has been used in training SVM models (Dalal & Triggs (2005)), boosted decision trees (Dosovitskiy et al. (2016)), shallow neural networks (Rowley et al. (1998)) and deep neural networks (Shrivastava et al. (2016)). Peng et al. (2018) and Wang et al. (2017a) considered generating hard samples by adversarially updating ranges of predefined transformations, such as occluding (Peng et al. (2018); Wang et al. (2017a)), scaling and rotating (Peng et al. (2018)). In these methods, flexible and complex transformation models cannot be used because adversarial update on these models may generate adversarial examples (Peng et al. (2018)). Adversarial examples are perturbed or transformed samples by a small amount, but they result in the classification model predicting an incorrect answer with high confidence (Szegedy et al. (2014)). Training classification model with adversarial examples provides the robustness on adversarial examples but often degenerate performance on clean test samples (Tramèr et al. (2018)). Recent studies have shown that adversarial updates in convolution based transformations (Baluja & Fischer (2018)) and spatial transformations (Xiao et al. (2018)) may generate adversarial examples.

**Supervised methods**  Lemley et al. (2017) designed an augmentation model that learns how to augment samples while learning a classification model in a way that reduces the training classification loss, but its effect on performance over test samples can only be empirically evaluated and cannot be predicted. Cubuk et al. (2018) considered small child models to compute validation loss to evaluate several augmentation policies over predefined transformations; however, learning requires

---

[1] Unsupervised methods may observe validation loss; however, they are referred to as unsupervised because the augmentation model is learned without label supervision.

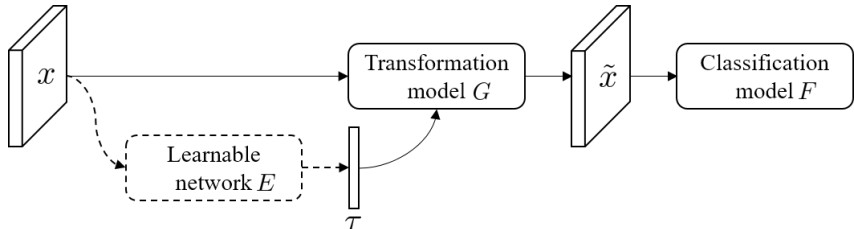

Figure 1: A generic data augmentation framework for the classification task. An input training sample $x$ is transformed to $\tilde{x}$ by a transformation model which is parameterized by $\tau$. The conventional method usually defines $G$ as a composition of predefined transformations with randomly sampled corresponding ranges $\tau$ (solid line path). In this paper, $G$ is a differentiable parametric model, and a learnable network $E$ for estimating $\tau$ from $x$ is also considered to obtain the transformed sample $\tilde{x}$ that maximizes generalization of the classification model (dashed line path).

a reinforcement learning framework because a gradient cannot flow from the validation loss to training samples, and non-differentiable predefined transformations stop the gradient. Furthermore, the validation loss of the small child model may be different from the validation loss of the final classification model.

## 2.2 GENERATIVE ADVERSARIAL NETWORKS

Goodfellow et al. (2014) proposed a generative adversarial network (GAN) that is a framework for training a deep generative network in an adversarial process. The framework considers the simultaneous training of two networks: a generator that generates a sample from random noise and a discriminator that estimates the probability that a sample came from the training data rather than the generator. The adversarial process is formed by a minimax two-player game between a discriminator that learns to distinguish a source of the sample and a generator that is trained to generate an indistinguishable sample by the discriminator.

Based on the GAN framework, several studies have shown their potential to use on data augmentation by generating class-conditional images (Odena et al. (2017); Nguyen et al. (2017); Mirza & Osindero (2014)) or improving the realism of synthetic samples (Shrivastava et al. (2017); Sixt et al. (2018); Wang et al. (2017b)); however, difficulties in learning generative models (relative to a classification model) require additional unlabeled (Odena et al. (2017); Nguyen et al. (2017); Mirza & Osindero (2014)) or synthetic (Shrivastava et al. (2017); Sixt et al. (2018); Wang et al. (2017b)) samples to improve performance.

## 2.3 INFLUENCE FUNCTIONS

The influence function is a classic technique from robust statistics (Cook & Weisberg (1980)) that estimates how the model parameters change due to up-weighting a particular training sample by a small amount. Cook & Weisberg (1980) focused on removing training points from linear models and Cook (1986); Thomas & Cook (1990); Wei et al. (1998) studied them with a wider variety of perturbations. Koh & Liang (2017) scaled up the influence function to non-convex and highly-complex models, including deep neural networks, by using efficient approximations based on Hessian-vector products (HVPs; Pearlmutter (1994)), conjugate gradients (Martens (2010)) and stochastic estimation (Agarwal et al. (2016)). They also considered the influence on the validation loss as up-weighting a particular training sample that can be computed without a retraining process and can be used for multiple purposes: debugging models, detecting dataset errors and even creating training-set attacks.

## 3 AUGMENTED DATA EVALUATION

Tuning or learning an augmentation model involves evaluating an augmented sample (Figure 1). Given the classification model, this is often performed by computing the training loss of the sample; however, the effect on test samples cannot be predicted and can only be computed in a trial and error

manner. Evaluating an augmented sample using validation loss requires learning two classifiers (including and excluding the sample during the learning process), which is computationally expensive because the models need to be fully trained and because of the cost to compute the loss over validation samples. Rather than repeat this prohibitive process, we propose a method to approximate the validation loss change due to a particular augmented sample without the retraining process.

## 3.1 PROBLEM SET UP

In a classification task, given an input space $\mathcal{X}$, an output space $\mathcal{Y}$ and parameter space $\Theta$, a learner aims to learn a classification model $F$ that maps $\mathcal{X} \mapsto \mathcal{Y}$ and is parameterized by $\theta$. Given a sample $z = (x, y) \in \mathcal{X} \times \mathcal{Y}$ and parameters $\theta \in \Theta$, let $l(z, \theta)$ be the loss evaluated on the sample $z$ at parameters $\theta$. With training data $\mathbf{z}^{\text{tr}} = \{z_i\}_{i=1}^N$, an empirical risk and an empirical risk minimizer are given as follows:

$$\mathcal{L}(\mathbf{z}^{\text{tr}}, \hat{\theta}(\mathbf{z}^{\text{tr}})) = \frac{1}{N} \sum_{i:z_i \in \mathbf{z}^{\text{tr}}} l(z_i, \hat{\theta}(\mathbf{z}^{\text{tr}})), \tag{1}$$

$$\text{where} \quad \hat{\theta}(\mathbf{z}^{\text{tr}}) = \arg\min_{\theta \in \Theta} \mathcal{L}(\mathbf{z}^{\text{tr}}, \theta). \tag{2}$$

To measure the generalization of the classification model $F$, validation data $\mathbf{z}^{\text{val}} = \{z_j\}_{j=1}^M$ is often considered, and generalization is approximated by an average of loss values over validation data $\mathbf{z}^{\text{val}}$ at parameter $\hat{\theta}(\mathbf{z}^{\text{tr}})$:

$$\mathcal{L}(\mathbf{z}^{\text{val}}, \hat{\theta}(\mathbf{z}^{\text{tr}})) = \frac{1}{M} \sum_{j:z_j \in \mathbf{z}^{\text{val}}} l(z_j, \hat{\theta}(\mathbf{z}^{\text{tr}})). \tag{3}$$

As shown in Figure 1, consider a label preserving transformation $G$ that maps $\mathcal{X} \mapsto \mathcal{X}$, and let $\tau$ be the control parameters, $\tilde{x} = G(x, \tau)$ be an augmented input, $\tilde{z} = (\tilde{x}, y)$ be an augmented sample, and $\tilde{\mathbf{z}}^{\text{tr}} = \{\tilde{z}_i\}_{i=1}^N$ be an augmented training dataset. In addition, consider a learnable network $E$ parameterized by $\phi$. It estimates control parameters $\tau$ given the input $x$. Thus, $\tilde{x} = G(x, E(x, \phi))$.

Given the transformation model $G$, the goal is then to find the optimal $\tau$ for each input $x$—otherwise, to find optimal parameters $\phi$ of $E$—that minimizes a classification loss over the validation data $\mathbf{z}^{\text{val}}$ when the classification model is learned using $\tilde{\mathbf{z}}^{\text{tr}}$:

$$\phi = \arg\min_{\phi \in \Phi} \mathcal{L}(\mathbf{z}^{\text{val}}, \hat{\theta}(\tilde{\mathbf{z}}^{\text{tr}})), \tag{4}$$

$$\hat{\theta}(\tilde{\mathbf{z}}^{\text{tr}}) = \arg\min_{\theta \in \Theta} \mathcal{L}(\tilde{\mathbf{z}}^{\text{tr}}, \theta). \tag{5}$$

Here, $\tilde{\mathbf{z}}^{\text{tr}} = \{\tilde{x}, y\}_{i=1}^N = \{G(x, E(x, \phi)), y\}_{i=1}^N$. Solving Equations 4–5 requires a bi-level optimization process such that Equation 4 can only be solved after Equation 5 is optimized.

## 3.2 INFLUENCE BY UPWEIGHTING A TRAINING SAMPLE

Consider a change in model parameters $\theta$ due to removing a particular training sample $z_i$. Formally, this change is $\hat{\theta}(\mathbf{z}^{\text{tr}} \backslash z_i) - \hat{\theta}(\mathbf{z}^{\text{tr}})$. Influence functions (Cook & Weisberg (1980); Koh & Liang (2017)) provide an efficient approximation without a retraining process to obtain $\hat{\theta}(\mathbf{z}^{\text{tr}} \backslash z_i)$. Let us consider the change in model parameters due to upweighting $z_i$ by an amount of $\epsilon l(z_i, \theta)$ in the loss function:

$$\hat{\theta}(\mathbf{z}^{\text{tr}} \cup \epsilon z_i) = \arg\min_{\theta \in \Theta} \mathcal{L}(\mathbf{z}^{\text{tr}}, \theta) + \epsilon l(z_i, \theta). \tag{6}$$

Then, from Cook & Weisberg (1980), the following approximation can be derived:

$$-\frac{1}{N} \mathcal{I}_{\text{up, params}}(z_i) \simeq \hat{\theta}(\mathbf{z}^{\text{tr}} \backslash z_i) - \hat{\theta}(\mathbf{z}^{\text{tr}}), \tag{7}$$

where

$$\mathcal{I}_{\text{up, params}}(z_i) \triangleq \left. \frac{d\hat{\theta}(\mathbf{z}^{\text{tr}} \cup \epsilon z_i)}{d\epsilon} \right|_{\epsilon=0} \tag{8}$$

$$= -H(\hat{\theta}(\mathbf{z}^{\text{tr}}))^{-1} \nabla_\theta l(z_i, \hat{\theta}(\mathbf{z}^{\text{tr}})). \tag{9}$$

Here, $H(\theta) \triangleq \frac{1}{N}\sum_{i=1}^{N}\nabla_\theta^2 l(z_i, \theta)$ is the Hessian evaluated at $\theta$.

Using Equation 9 and applying the chain rule, the influence of upweighting $z_i \in \mathbf{z}^{\text{tr}}$ on the validation loss at $z_j \in \mathbf{z}^{\text{val}}$ can be approximated (Koh & Liang (2017)) as shown below:

$$-\frac{1}{N}\mathcal{I}_{\text{up, loss}}(z_i, z_j) \simeq l(z_j, \hat{\theta}(\mathbf{z}^{\text{tr}}\backslash z_i)) - l(z_j, \hat{\theta}(\mathbf{z}^{\text{tr}})), \tag{10}$$

where

$$\mathcal{I}_{\text{up, loss}}(z_i, z_j) \triangleq \left.\frac{dl(z_j, \hat{\theta}(\mathbf{z}^{\text{tr}} \cup \epsilon z_i))}{d\epsilon}\right|_{\epsilon=0} \tag{11}$$

$$= \nabla_\theta l(z_j, \hat{\theta}(\mathbf{z}^{\text{tr}}))^\top \left.\frac{d\hat{\theta}(\mathbf{z}^n \cup \epsilon z_i)}{d\epsilon}\right|_{\epsilon=0} \tag{12}$$

$$= -\nabla_\theta l(z_j, \hat{\theta}(\mathbf{z}^{\text{tr}}))^\top H(\hat{\theta}(\mathbf{z}^{\text{tr}}))^{-1}\nabla_\theta l(z_i, \hat{\theta}(\mathbf{z}^{\text{tr}})). \tag{13}$$

For the validation dataset $\mathbf{z}^{\text{val}}$, Equation 12 is simply expanded by:

$$\mathcal{I}_{\text{up, loss}}(z_i, \mathbf{z}^{\text{val}}) = -\nabla_\theta \mathcal{L}(\mathbf{z}^{\text{val}}, \hat{\theta}(\mathbf{z}^{\text{tr}}))^\top H(\hat{\theta}(\mathbf{z}^{\text{tr}}))^{-1}\nabla_\theta l(z_i, \hat{\theta}(\mathbf{z}^{\text{tr}})). \tag{14}$$

Equation 11 describes a gradient of $l(z_j, \hat{\theta}(\mathbf{z}^{\text{tr}}))$ w. r. t. $\epsilon$ at nearby $\epsilon = 0$, and the influence of removing $z_i$ can be approximate by Equation 10.

### 3.3 Influence by augmentation

With a training sample $z_i$ and the corresponding augmented training sample $\tilde{z}_i$, let $\hat{\theta}(\mathbf{z}^{\text{tr}} \cup \epsilon\tilde{z}_i\backslash\epsilon z_i)$ be the estimate of $\theta$ with downweighting $z_i$ and with upweighting $\tilde{z}_i$ by the $\epsilon$ amount, and let $\hat{\theta}(\mathbf{z}^{\text{tr}} \cup \tilde{z}_i\backslash z_i)$ be the estimate of $\theta$ by replacing $z_i$ with $\tilde{z}_i$. An analogous approximation of Equations 10–13 yields:

$$-\frac{1}{n}\mathcal{I}_{\text{aug, loss}}(z_i, \mathbf{z}^{\text{val}}) \simeq \mathcal{L}(\mathbf{z}^{\text{val}}, \hat{\theta}(\mathbf{z}^{\text{tr}})) - \mathcal{L}(\mathbf{z}^{\text{val}}, \hat{\theta}(\mathbf{z}^{\text{tr}} \cup \tilde{z}_i\backslash z_i)), \tag{15}$$

where the influence function $\mathcal{I}_{\text{aug, loss}}(z_i, \tilde{z}_i, \mathbf{z}^{\text{val}})$ is:

$$\mathcal{I}_{\text{aug, loss}}(z_i, \tilde{z}_i, \mathbf{z}^{\text{val}}) \triangleq \left.\frac{d\mathcal{L}(\mathbf{z}^{\text{val}}, \hat{\theta}(\mathbf{z}^{\text{tr}} \cup \epsilon\tilde{z}_i\backslash\epsilon z_i))}{d\epsilon}\right|_{\epsilon=0} \tag{16}$$

$$= \nabla_\theta \mathcal{L}(\mathbf{z}^{\text{val}}, \hat{\theta}(\mathbf{z}^{\text{tr}}))^\top \left.\frac{d\hat{\theta}(\mathbf{z}^{\text{tr}} \cup \epsilon\tilde{z}_i\backslash\epsilon z_i)}{d\epsilon}\right|_{\epsilon=0} \tag{17}$$

$$= -\nabla_\theta \mathcal{L}(\mathbf{z}^{\text{val}}, \hat{\theta}(\mathbf{z}^{\text{tr}}))^\top H(\hat{\theta}(\mathbf{z}^{\text{tr}}))^{-1}\left(\nabla_\theta l(\tilde{z}_i, \hat{\theta}(\mathbf{z}^{\text{tr}})) - \nabla_\theta l(z_i, \hat{\theta}(\mathbf{z}^{\text{tr}}))\right) \tag{18}$$

$$= \mathcal{I}_{\text{up, loss}}(\tilde{z}_i, \mathbf{z}^{\text{val}}) - \mathcal{I}_{\text{up, loss}}(z_i, \mathbf{z}^{\text{val}}). \tag{19}$$

The influence function $\mathcal{I}_{\text{aug, loss}}(z_i, \tilde{z}_i, \mathbf{z}^{\text{val}})$ predicts the difference in influences of the augmented sample and the original sample. In addition, the influence function can be used to evaluate the effectiveness of the augmented data $\tilde{z}_i$ in comparison to its original. Because the augmentation model is deterministic, this influence by replacement reflects the augmentation process during learning final classification model; however, it does not mean that original samples are excluded during learning the classifier. If the original sample has the most influence than other transformed samples, then $E$ learns identity transformation for that sample.

### 3.4 Reformulation of influence functions

To compute influence functions efficiently, we adopt techniques such as conjugate gradients and stochastic estimation used in Koh & Liang (2017). Given $G$ and $F$, Hessian-vector products (HVPs) $s_{\text{val}} = H(\hat{\theta}(\mathbf{z}^{\text{tr}}))^{-1}\nabla_\theta \mathcal{L}(\mathbf{z}^{\text{val}}, \hat{\theta}(\mathbf{z}^{\text{tr}}))$ is precomputed and cumulated for validation samples using conjugate gradients and stochastic estimation techniques. The HVPs are fixed during learning $E$ and are used to compute influence functions of augmented samples. We further approximate the influence function by only considering the top fully connected layer of $F^2$, thus the remaining $\nabla_\theta l(z_i, \hat{\theta}(\mathbf{z}^{\text{tr}}))$ can be represented by a simple closed form. This makes it possible to represent $s_{\text{val}}\nabla_\theta l(z_i, \hat{\theta}(\mathbf{z}^{\text{tr}}))$ as a closed form by regarding $s_{\text{val}}$ as a fixed vector. The gradient from this flows through fixed $F$ and $G$, and is then used to update $E$ by the chain rule.

---

[2] This approximation is only considered during learning $E$. All parameters of $F$ are updated during learning $F$

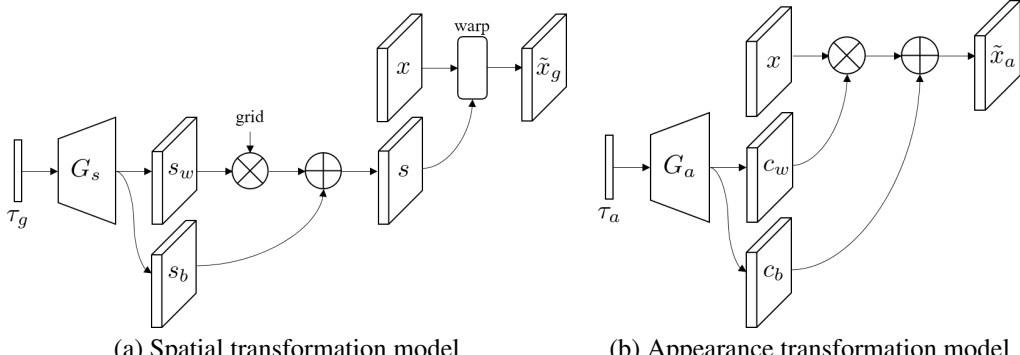

(a) Spatial transformation model         (b) Appearance transformation model

Figure 2: The proposed transformation models for spatial and appearance transformation. (a) Spatial transformation model $G_s$ takes an input $\tau_g$ and then generates $s_w$ and $s_b$. Flow field $s$ denotes the coordinates to be transformed and is obtained by operations like the point-wise affine transform by $s_w$ and $s_b$. The warp operation transforms $x$ to $\tilde{x}$ by interpolating $x$ based on the coordinates in $s$. (b) Appearance transformation model $G_a$ takes an input $\tau_a$ and then generates $c_w$ and $c_b$. $\tilde{x}$ is obtained by filtering $x$ by a filter with weights $c_w$ and bias $c_b$.

## 4 TRANSFORMATION MODELS

Transformations for augmenting images can be categorized as either spatial transformations or appearance transformations. In this section, the proposed transformation models that generalize both transformations are described. The models are differentiable, so a gradient from the influence function can be propagated to $E$ during learning.

### 4.1 SPATIAL TRANSFORMATION MODEL

Spatial transformations include random flip, crop, scaling, rotation, shearing, translation, and affine transformation. These transformations can be defined by the coordinate change in pixel locations. The proposed transformation model for spatial transformation is illustrated in Figure 2-(a). We define spatial transformation—in a way similar to Jaderberg et al. (2015); Xiao et al. (2018)—by

$$\begin{bmatrix} s(i,j,1) \\ s(i,j,2) \end{bmatrix} = \begin{bmatrix} s_w(i,j,1) & s_w(i,j,2) \\ s_w(i,j,3) & s_w(i,j,4) \end{bmatrix} \begin{bmatrix} i \\ j \end{bmatrix} + \begin{bmatrix} s_b(i,j,1) \\ s_b(i,j,2) \end{bmatrix} + \begin{bmatrix} i \\ j \end{bmatrix}. \tag{20}$$

Here, $i, j$ denotes the source coordinate, $s_w(i, j)$, and $s_b(i, j)$ denote the multiplication factor and bias, respectively. Note that when we perform global average pooling on $s_w$ and $s_b$, the transformation is reduced to an affine transform. In this formulation, the spatial transformation is fully defined by $s_w$ and $s_b$ and these parameters are what we want to generate from $\tau_g$.

Spatial transformation model $G_g$ takes an input $\tau_g$ and then generates $s_w$ and $s_b$. Flow field $s$ denotes the coordinates to be transformed and is obtained by operations like the point-wise affine transform by $s_w$ and $s_b$. The warp operation in Figure 2-(a) transforms $x$ to $\tilde{x}$ based on the bilinear interpolation indexed by $s$. The model is designed by a stacked transposed convolutional network with its final layer having $4 + 2$ channels. After we obtain $s$, average pooling is applied to smoothing $s$, and then the image is warped by bilinear interpolation, which is differentiable (Jaderberg et al. (2015)). All computations in this formulation can be implemented by a feed-forward neural network.

### 4.2 APPEARANCE TRANSFORMATION MODEL

Transformations in appearance include enhance contrast, brightness, color and hue shift. These transformations can be formed by $1 \times 1$ spatial dimension filters. Thus, to formulate the appearance transformation model, we focused on generating $1 \times 1$ spatial filters. The proposed transformation model for the appearance transformation is illustrated in Figure 2-(b). The appearance transformation model $G_a$ takes an input $\tau_a$ and then generates $c_w$ and $c_b$. $c_w$ and $c_b$ are average pooled as the spatial transformation model. Transformed image $\tilde{x}$ is obtained by $x + \delta x$, where $\delta x$ is obtained by

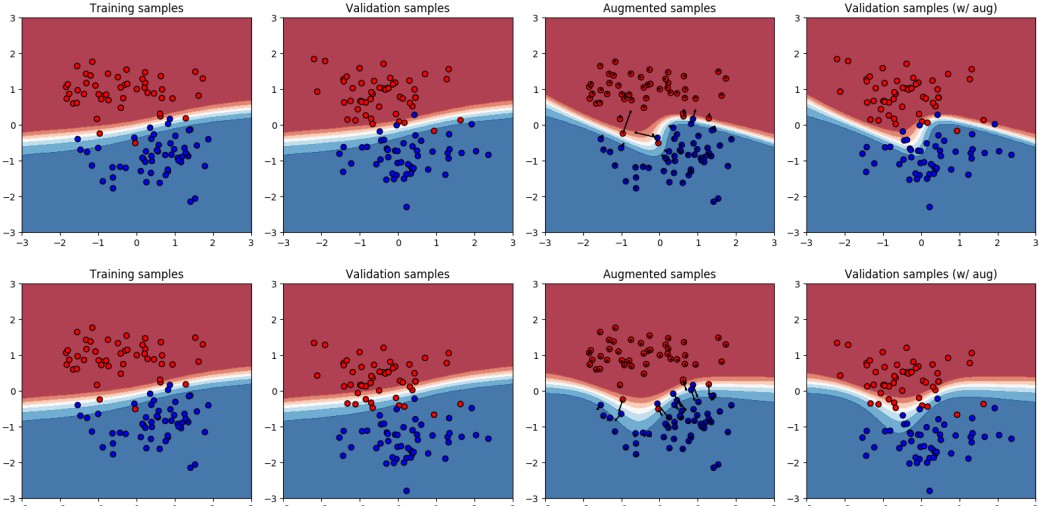

Figure 3: Toy example on a synthetic binary classification dataset. Red and blue dots represent data points of class red and blue. A contour plot of predictive probability is overlaid. Each column represents: (i) training samples and the predictive probability contour plot from model parameters learned using training samples, (ii) validation samples and the predictive probability contour plot from model parameters learned using training sample, (iii) augmented samples to maximize influence and the predictive probability contour plot from model parameters learned using augmented samples, (iv) validation samples with the predictive probability contour plot from model parameters learned using augmented samples.

filtering $x$ with filter weights $c_w$ and bias $c_b$. The model is designed by a transposed convolutional network with its final layer having $1 \times 1 + 1$ channels for grey images and $3 \times 3 + 3$ channels for RGB images.

## 4.3 LEARNING TRANSFORMATION MODELS

During learning, $G$ is trained based on the GAN framework and is fixed thereafter. Then $E$ is trained to predict $\tau$, which maximizes the influence function for each $x$ and $\tilde{x} = G(x, E(x))$ pair. In practice, we combine the spatial and appearance transformation model by concatenating $(\tau_s, \tau_a)$ and $(s_w, s_b, c_w, c_b)$. The WGAN-GP (Gulrajani et al.) is used for training $G$ with modifications. First, two discriminators over the image space and flow field $s$ space are considered. The discriminator over the image space is updated to distinguish $\tilde{x}$ and $x$, and the discriminator over flow field $s$ space is updated to distinguish $s = G_s(\tau)$ and manually generated flow fields that will result in random affine transforms. Moreover, to prevent the identity transformation, the inverse of the mean-square error on the flow field space and image space are additionally considered when updating $G$. For the details of hyperparameters and architectures, refer to Appendix A and B.

## 5 EXPERIMENTS

### 5.1 TOY 2D DATASET

To visualize the proposed method, we conduct experiments on a toy 2D dataset. Transitions along the x-axis and y-axis are considered to the augmentation model, and the influence is computed by validation samples. A simple 3-layer neural network is considered for classification. In Figure 3, red and blue dots represent data points of class red and blue, and a contour plot of predictive probability is overlaid.

In the first row of Figure 3, the influence value of augmented samples near the mis-classified validation sample is high, so the augmentation model learns transitions toward the mis-classified validation

samples. In the second row of Figure 3, we generate training and validation data and then move the validation data to the downside to make a mismatch with the training data. In this setting, the influence value of the downside transition is high, so the augmentation model learns downside transition.

Table 1: MNIST accuracies of various data augmentation methods.

| Method | The number of labeled data | |
|---|---|---|
| | 1% | 10% |
| None | 90.2% | 97.3% |
| Heur. | 95.9% | 99.0% |
| Ratner (MF) | 96.5% | 99.2% |
| Ratner (LSTM) | 96.7% | 99.1% |
| Proposed | 96.7% | 99.3% |

Table 2: CIFAR-10 accuracies of various data augmentation methods.

| Method | The number of labeled data | |
|---|---|---|
| | 10% | 100% |
| None | 66.0% | 87.8% |
| Heur. | 77.5% | 92.3% |
| Ratner (MF) | 79.8% | 94.4% |
| Ratner (LSTM) | 81.5% | 94.0% |
| Proposed | 82.1% | 94.8% |

## 5.2 MNIST DATASET

We ran experiments on the MNIST using only a subset of the class labels to train the classification models and treating the rest as unlabeled data. We used a four-layer all-convolutional CNN for classification. Experiments were conducted under the same setting as Ratner et al. (2017). In Table 1, we list classification accuracies for various data augmentation methods from Ratner et al. (2017). *None* indicates that no augmentation is applied, and *Heur* is the standard heuristic approach of applying random compositions of the given set of transformation operations. *Ratner* denotes the results from Ratner et al. (2017). We used a 128 dimensional $\tau$ and four-layered convolutional neural network with additional fully connected layer for $E$ and a symmetry transposed convolutional neural network for $G$. For the proposed method, we additionally use the random cropping technique as in Ratner et al. (2017). For the details, refer to Appendix A and B.

## 5.3 CIFAR-10 DATASETS

We ran experiments on the CIFAR-10 using a subset of the class labels to train the classification models and treating the rest as unlabeled data. We used a ResNet-56 (He et al. (2016)). In Table 2, we list classification accuracies for various data augmentation methods from Ratner et al. (2017). We used a 128 dimensional $\tau$ and four-layered convolutional neural network with additional fully connected layer for $E$ and a symmetry transposed convolutional neural network for $G$. For the proposed method, we additionally use the random cropping and horizontal flip technique as in Ratner et al. (2017). For the details, refer to Appendix A and B.

## 6 CONCLUSION

Data augmentation is a technique for avoiding overfitting and improving generalization by increasing the size of labeled datasets; however, it is currently conducted in a trial and error manner. Composition of predefined transformations, such as rotation, scaling and cropping, are performed on training samples, and its effect on performance over test samples can only be empirically evaluated and cannot be predicted. This paper considered the influence function, which predicts how generalization in terms of a validation loss is affected by a particular augmented training sample without comparing the performances that include and exclude it in the training process. We also proposed a differentiable augmentation model that generalizes the conventional composition of predefined transformations. The differentiable augmentation model and reformulation of the influence function allowed the augmented model parameters to be updated by backpropagation to minimize the validation loss. Our results confirmed that the proposed method provides better generalization than conventional data augmentation methods.

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

## APPENDIX

## A   ARCHITECTURES

### A.1   TRANSFORMATION MODEL DESCRIPTION FOR MNIST

| Module | Layer | Layer description |
|---|---|---|
| | Input | $x \in \mathbb{R}^{28 \times 28 \times 1}$ |
| | Conv2d | $4 \times 4$, stride 1, $1 \to 16$, ReLU |
| | Conv2d | $4 \times 4$, stride 2, $16 \to 32$, ReLU |
| | Conv2d | $4 \times 4$, stride 2, $32 \to 64$, ReLU |
| $E$ | Reshape | $7 \times 7 \times 64 \to 3136$ |
| | Linear | $3136 \to 128$, Tanh |
| | Output | $\tau \in \mathbb{R}^{128}$ |
| | Input | $\tau \in \mathbb{R}^{128}$ |
| | Linear | $128 \to 3136$ |
| | Reshape | $3136 \to 7 \times 7 \times 64$ |
| | ConvTranspose2d | $4 \times 4$, stride 2, $64 \to 32$, ReLU |
| $G$ | ConvTranspose2d | $4 \times 4$, stride 2, $32 \to 16$, ReLU |
| | ConvTranspose2d | $4 \times 4$, stride 1, $16 \to 8$, Identity |
| | Output | $s \in \mathbb{R}^{28 \times 28 \times 8}$ |
| | Input | $x \in \mathbb{R}^{28 \times 28 \times 1}$ |
| | Conv2d | $4 \times 4$, stride 1, $1 \to 16$, LeackyReLU 0.2 |
| | Conv2d | $4 \times 4$, stride 2, $16 \to 32$, LeackyReLU 0.2 |
| | Conv2d | $4 \times 4$, stride 2, $32 \to 64$, LeackyReLU 0.2 |
| $D$ | Reshape | $7 \times 7 \times 64 \to 3136$ |
| | Linear | $3136 \to 1$, |
| | Output | $D(x) \in \mathbb{R}^1$ |

### A.2   TRANSFORMATION MODEL DESCRIPTION FOR CIFAR-10

| Module | Layer | Layer description |
|---|---|---|
| | Input | $x \in \mathbb{R}^{32 \times 32 \times 3}$ |
| | Conv2d | $4 \times 4$, stride 1, $1 \to 16$, ReLU |
| | Conv2d | $4 \times 4$, stride 2, $16 \to 32$, ReLU |
| | Conv2d | $4 \times 4$, stride 2, $32 \to 64$, ReLU |
| $E$ | Conv2d | $4 \times 4$, stride 2, $64 \to 128$, ReLU |
| | Reshape | $4 \times 4 \times 128 \to 2048$ |
| | Linear | $2048 \to 128$, Tanh |
| | Output | $\tau \in \mathbb{R}^{128}$ |
| | Input | $\tau \in \mathbb{R}^{128}$ |
| | Linear | $128 \to 2048$ |
| | Reshape | $2048 \to 4 \times 4 \times 128$ |
| | ConvTranspose2d | $4 \times 4$, stride 2, $128 \to 64$, ReLU |
| $G$ | ConvTranspose2d | $4 \times 4$, stride 2, $64 \to 32$, ReLU |
| | ConvTranspose2d | $4 \times 4$, stride 2, $32 \to 16$, ReLU |
| | ConvTranspose2d | $4 \times 4$, stride 1, $16 \to 18$, Identity |
| | Output | $s \in \mathbb{R}^{32 \times 32 \times 18}$ |
| | Input | $x \in \mathbb{R}^{28 \times 28 \times 1}$ |
| | Conv2d | $4 \times 4$, stride 1, $1 \to 16$, LeackyReLU 0.2 |
| | Conv2d | $4 \times 4$, stride 2, $16 \to 32$, LeackyReLU 0.2 |
| | Conv2d | $4 \times 4$, stride 2, $32 \to 64$, LeackyReLU 0.2 |
| $D$ | Conv2d | $4 \times 4$, stride 2, $64 \to 128$, LeackyReLU 0.2 |
| | Reshape | $4 \times 4 \times 128 \to 2048$ |
| | Linear | $2048 \to 1$, |
| | Output | $D(x) \in \mathbb{R}^1$ |

## B  OPTIMIZERS

### B.1  MNIST

| Module | Optimizer |
|--------|-----------|
| G, D | ADAM, lr=0.0001, $\beta_1$=0.5, $\beta_2$=0.9, batch size=128 |
| E | ADAM, lr=0.01, $\beta_1$=0.9, $\beta_2$=0.99, batch size=128 |
| F | ADAM, lr=0.01, $\beta_1$=0.9, $\beta_2$=0.99, batch size=128 |

### B.2  CIFAR-10

| Module | Optimizer |
|--------|-----------|
| G, D | ADAM, lr=0.0001, $\beta_1$=0.5, $\beta_2$=0.9, batch size=128 |
| E | ADAM, lr=0.01, $\beta_1$=0.9, $\beta_2$=0.99, batch size=128 |
| F | Same setting as He et al. (2016) |

## C  INTERPOLATING IN $\tau$ SPACE

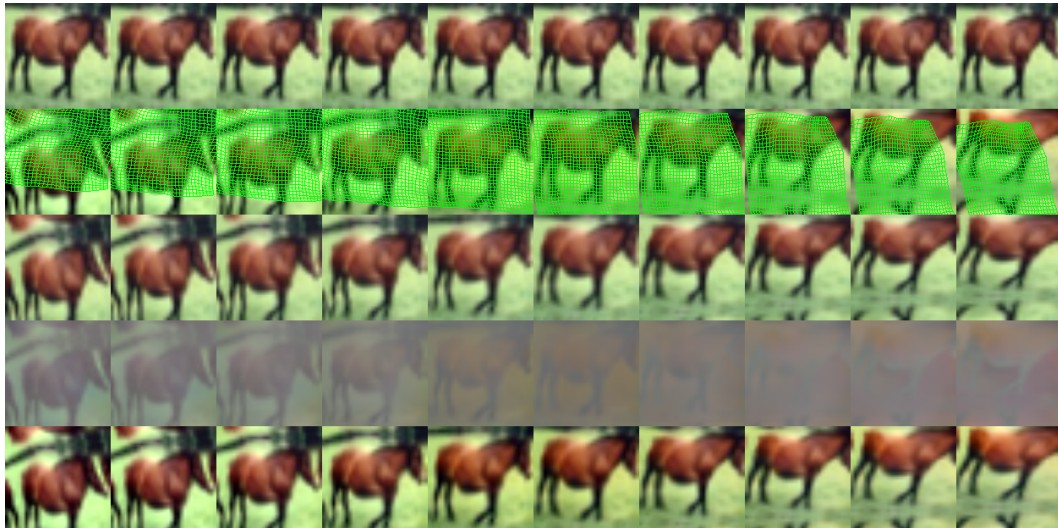

Figure 4: Images are transformed by applying linearly interpolated ten $\tau$'s to the same training sample in CIFAR-10 dataset. Each row represents: (i) the training image, (ii) spatial transformation model outputs, (iii) spatially transformed images, (iv) appearance transformation model outputs and (v) final transformed images.

