# OpenReview forum: "Learning to Augment Influential Data"
_ICLR.cc/2019/Conference_

### Official Review · AnonReviewer2 · 2018-11-04

**Rating:** 5
**Confidence:** 4

**Review:**

This paper proposed an


1. For me, the argument of the paper is ambitious. Data augmentation for DNN includes different perspective, including nonlinearity, adversarial etc. Generalization of  spatial and appearance models is not enough. The model formulate from a simple classification setting but does not involve too many for DNN models.  I put more references below.

2. The experimental results are not strong. Not all strong baselines are included (I put some in the references). The improvements are marginal. Besides, I need more experimental setting information.

3. The writing is not clear. For the related work part, it included many paragraph which are not related to the work, (e.g. GANs). In the introduction part, it did not mention the generalization of both spatial and appearance models, which is the main contribution.

References:
a. Good Semi-supervised Learning that Requires a Bad GAN
b. Semi-supervised Learning with GANs: Manifold Invariance with Improved Inference
c. Temporal ensembling for semi-supervised learning

---

> ### Author Response · Authors · 2018-11-26
> **Responses to AnonReviewer2**
>
> The authors would like to thank all the reviewers for their valuable comments. There seems to be a gap between what was intended to be conveyed by the manuscript and what was understood by the reviewer. Hopefully, the revised manuscript will allow the reviewer to have a better understanding of what was intended.
>
> [R] For me, the argument of the paper is ambitious. Data augmentation for DNN includes different perspective, including nonlinearity, adversarial etc. Generalization of spatial and appearance models is not enough. The model formulate from a simple classification setting but does not involve too many for DNN models. I put more references below.
> [A] The proposed transformation model generalizes non-rigid (nonlinear data augmentation is a rather ambiguous term-the reviewers are probably referring to non-rigid warping) warping. The adversarial method is limited in their flexibility of transformation as described in Section 2.1. This is because the adversarial update on flexible and complex transformation model yields an adversarial example. The paper only considered the classification setting, but extending it to other tasks is straightforward given the differentiable objectives (Eqs 14 and 19 are not restricted to classification setting).
>
> [R] The experimental results are not strong. Not all strong baselines are included (I put some in the references). The improvements are marginal. Besides, I need more experimental setting information.
> [A] Thank you for the baselines of (semi-)supervised learning algorithms; however, the paper is not about the semi-supervised learning but is about the data augmentation. The proposed augmentation method can be used in conjunction with methods proposed in [a,b,c]. For example, [c] reported the results with or without augmentation in Tables 1 and 2. The benefit of the proposed method can be investigated by measuring the performance gain of the proposed augmentation method in comparison with the conventional augmentation method or without augmentation given the algorithm in [c].
>
> [R] The writing is not clear. For the related work part, it included many paragraph which are not related to the work, (e.g. GANs). In the introduction part, it did not mention the generalization of both spatial and appearance models, which is the main contribution.
> [A] The manuscript has been revised by many people including a native English speaker to improve readability.
> As we stated in Section 2.2, we included GAN in related work to exploit its potential to augment data by generating class-conditional images.
> As AnonReviewer1 summarized, our main contribution is not the generalization of both spatial and appearance models, but our contribution is to propose an extension of the influence function for data augmentation. Influence of augmentation on validation loss is approximated and the augmentation model is learned under this approximation. Differentiable transformation models, the generalization of both spatial and appearance models, are proposed to carry out gradients of influence function to the transformation model. Also, we briefly mentioned the generalization of both spatial and appearance models by “We also propose a differentiable augmentation model that generalizes the conventional augmentation method by the composition of predefined transformations”.

---

### Official Review · AnonReviewer3 · 2018-11-05
**Interesting paper, Inspiring theory, but need more experiments**

**Rating:** 6
**Confidence:** 4

**Review:**

[Summary]

This paper proposes a differentiable framework to learn to augment data for image classification. In particular, it uses spatial transformer and GANs as parametric data augmenters, and it formulates the validation set loss with respect to the data augmenter in a differentiable manner.

[Pros]

1.	The proposed method does not require many trials of model training under different training data, and it learns the data augmentation directly using the final classification objective.
2.	It is inspiring to extend the differentiable form of influence function across the training and validation set and then across the original and augmented data. This paper also makes use of the most recent related advance to enable stochastic learning. The theory of the paper is nice.
3.	The experimental results on MNIST (with less labeled data) and Cifar-10 are encouraging.

[Cons]

1.	Experimental results can be stronger. Especially when compared to Ratner et al., this proposed method results in marginal performance gain. Given that Ratner et al.’s method trained the data augmentation module without supervision, the supervised learning in this paper does not show strong results. In addition, the paper did not report results on a more practical dataset (such ImageNet and Places). Even for Cifar-10, the reported numbers are away from the state-of-the-art. It is important to show the practical significance of the proposed method.
2.	Data augmentation is naturally expected to be random, but the proposed method seems to learn a deterministic parameter for the augmenting transformation, which looks unnatural and limited. (Please clarify if I missed anything.)
3.	The proposed method requires a parametric model (e.g, STN, GAN). However, differentiable parametric models are not always easy to design. This probably can be the biggest obstacle to apply the proposed method widely.

Overall, the proposed method is very interesting. However, the experimental results are limited, and more discussions are needed.

---

> ### Author Response · Authors · 2018-11-26
> **Responses to AnonReviewer3**
>
> We appreciate you for the detailed comments. We would like to share our responses to the concerns raised by the reviewer.
>
> [R] Experimental results can be stronger. Especially when compared to Ratner et al., this proposed method results in marginal performance gain. Given that Ratner et al.’s method trained the data augmentation module without supervision, the supervised learning in this paper does not show strong results. In addition, the paper did not report results on a more practical dataset (such ImageNet and Places). Even for Cifar-10, the reported numbers are away from the state-of-the-art. It is important to show the practical significance of the proposed method.
> [A] We would like to point out that significance of supervision during learning an augmentation module is different from that of during learning a classification module. Supervision of the classification module (classical supervised learning) is expected to show dramatic performance gain over that of without supervision (classical unsupervised learning); however, supervision of the augmentation module may not show significant performance gain because the performance is measured by the end classifier which already includes label supervision.
> The experimental results focused on comparisons between the proposed method, heuristic and Ratner et al. under the exact same setting (hyperparameters and an architecture of the end classifier); however, we agree with the comment and were working on the experiments on ImageNet. But single run of the end classification model take roughly one week on ImageNet. Alternatively, we are currently working on the experiments on Cifar-10 for various state-of-the-art architectures and will report the results as soon as possible.
>
> [R] Data augmentation is naturally expected to be random, but the proposed method seems to learn a deterministic parameter for the augmenting transformation, which looks unnatural and limited. (Please clarify if I missed anything.)
> [A] As described in the manuscript, the experiments were conducted in the same setting as Ratner et al.: training images are randomly transformed (crop and flip) before they fed into the transformation model. This injects randomness to augmented images. We revised the manuscript to include this process for clarity.
>
> [R] The proposed method requires a parametric model (e.g, STN, GAN). However, differentiable parametric models are not always easy to design. This probably can be the biggest obstacle to apply the proposed method widely.
> [A] In deep learning, to find stable setups of hyperparameters and an architecture is not easy, especially for GAN, as you pointed out. But we found that using the known stable setup (from the original WGAN-GP paper) is enough to learn transformation models. For detail, we used the same setup as the original WGAN-GP paper (see the revised paper Appendix A and B):
>     Optimizer: ADAM, lr=0.0001, beta_1=0.5, beta_2=0.9,
>     Generator: Stacked transposed convolutional neural networks with ReLU,
>     Discriminator: Stacked convolutional neural networks with LeakyReLU (0.2)
> Other GAN variants with their known stable setups also worked, e. g. DCGAN setup used in the original DCGAN paper (ADAM optimizer with lr=0.0002, beta_1=-.5, beta_2=0.999). Furthermore, learning of the proposed transformation model is not much sensitive to hyperparameters and architecture design. We believe that this is because of the difference in objectives between the GAN variants and the transformation model. The GAN variants generate images from noise; however, the transformation model generates transformation parameters to be applied to the given image, which has more simple patterns than the image itself.

---

> > ### Comment · AnonReviewer3 · 2018-12-17
> > **Thank you for the response**
> >
> > Thank you for the detailed response. Though I partially agree with the author's responses (e.g., I do not agree with the argument about randomness), I believe this paper shows enough values to be above the bar.

---

### Official Review · AnonReviewer1 · 2018-11-05
**valuable and publishable, some potential improvements**

**Rating:** 6
**Confidence:** 4

**Review:**

This paper proposes an extension of the influence function study of Koh and Liang (2017) to data augmentation.  Influence of augmentation, carried out via a parameterized and differentiable model, on validation loss is approximated and the augmentation model is learned under this approximation.  Overall I think it is a valuable and publishable contribution.  I do find the paper to be unclear and perhaps could be improved in a few ways.  My main comments are:

* The biggest question I have is it seems from Eq. 15 that authors are proposing an augmentation approach where the augmented samples replace the original samples and not co-exist with them in the training set.  I am not sure why Eq. 15 has to be set up like that, please elaborate.

* In Section 3.4 it is stated that only top fully connected layer of F is considered to compute influence function for augmentation.  Does this also mean that when F is updated on augmented data only the top layer is updated?  Please clarify.

* The paper is a bit difficult to follow due to lack of clarity and few errors:
     - Section 2.1, Adversarial methods, “In these methods, a simple composition…adversarial examples” sentence is unclear
     - Page 2 footnote “however, they are referred to as unsupervised due to learning is not involved” sentence is unclear
     - Section 3.3 \tilde{z} in first line should be \tilde{z_i}
     - Eq. 15 LHS should include \tilde{z_i}
     - Section 3.4 “adopts” -> “adopt”
     - Section 3.4 “HVP” used without defining

* Empirical evidence, while not extensive, is satisfactory.

---

> ### Author Response · Authors · 2018-11-26
> **Responses to AnonReviewer1**
>
> We thank you for the constructive comments on our work. We have revised the manuscript to better explain the points you mentioned, and we hope this improving the clarity of the paper.
>
> [R] The biggest question I have is it seems from Eq. 15 that authors are proposing an augmentation approach where the augmented samples replace the original samples and not co-exist with them in the training set. I am not sure why Eq. 15 has to be set up like that, please elaborate.
> [A] The augmentation process of an input image x is G(x, E(x)), and this is deterministic because E and G are neural networks. Thus, during learning the end classifier, the augmented sample actually replaces the original sample.
> We would like to point out that this replacement does not mean that original samples are excluded during learning the end classifier. If the original image x has the most influence than other transformed images, then E learns to generate G(x, E(x)) to be similar with x.
>
> [R] In Section 3.4 it is stated that only top fully connected layer of F is considered to compute influence function for augmentation. Does this also mean that when F is updated on augmented data only the top layer is updated? Please clarify.
> [A] The approximation is only considered during learning E. All parameters of F are updated during learning end classifier.
>
> [R] The paper is a bit difficult to follow due to lack of clarity and few errors.
> [A] We fixed all errors, and the manuscript has been revised for clarity and has been proofread by a native English speaker.

---

### Meta-Review · Area_Chair1 · 2018-12-15
**Interesting contribution, but not fully developed yet.**

**Confidence:** 4
**Recommendation:** Reject

**Metareview:**

This paper proposes and end-to-end trainable architecture for data augmentation, by defining a parametric model for data augmentation (using spatial transformers and GANs) and optimizing validation classification error through the notion of influence functions. Experiments are reported on MNIST and CIfar-10.

This is a borderline submission. Reviewers found the theoretical framework and problem setup to be solid and promising, but were also concerned about the experimental setup and the lack of clarity in the manuscript. In particular, one would like to evaluate this model against similar baselines (e.g. Ratner et al) on a large-scale classification problem. The AC, after taking these comments into account and making his/her own assessment, recommends rejection at this time, encouraging the authors to address the above comments and resubmit this promising work in the next conference cycle.